# Unknown-Aware Deep Neural Network

## Abstract

An important property of image classification systems in the real world is that they both accurately classify objects from target classes ("knowns") and safely reject unknown objects ("unknowns") that belong to classes not present in the training data. Unfortunately, although the strong generalization ability of existing CNNs ensures their accuracy when classifying known objects, it also causes them to often assign an unknown to a target class with high confidence. As a result, simply using low-confidence detections as a way to detect unknowns does not work well. In this work, we propose an Unknown-aware Deep Neural Network (UDN for short) to solve this challenging problem. The key idea of UDN is to enhance existing CNNs to support a *product* operation that models the *product relationship* among the features produced by convolutional layers. This way, missing a single key feature of a target class will greatly reduce the probability of assigning an object to this class. UDN uses a learned ensemble of these product operations, which allows it to balance the contradictory requirements of accurately classifying known objects and correctly rejecting unknowns. To further improve the performance of UDN at detecting unknowns, we propose an information-theoretic regularization strategy that incorporates the objective of rejecting unknowns into the learning process of UDN. We experiment on benchmark image datasets including MNIST, CIFAR-10, CIFAR-100, and SVHN, adding unknowns by injecting one dataset into another. Our results demonstrate that UDN consistently outperforms state-of-the-art methods at rejecting unknowns – up to 20 point gains in accuracy, while still preserving the classification accuracy.

## 1 Introduction

**Motivation.** In recent years, Convolutional Neural Networks (CNN) have been used with great success for a rich variety of classification problems, particularly when dealing with high dimensional, complex data such as images or time series (Goodfellow et al., 2016). A CNN classifier (Krizhevsky et al., 2012) typically classifies test objects as one of the *target classes* supplied in the training set. In this, state-of-the-art classifiers make the implicit assumption that all testing objects belong to one of the target classes. However, this assumption is rarely true in real-world deployments of CNN classifiers. Consider for example, an autonomous car or healthcare system: it is extremely likely that the system will be exposed to objects that were not in its training set. We call such objects "unknowns".

Clearly, blindly assigning these unknowns into one of the target classes degrades the prediction accuracy. Worst yet, it can lead to serious safety concerns. For example, in a collaboration with a top hospital in the US (name removed due to anonymity), we have been developing a seizure detector that classifies patients into different types of seizures based on EEG signals collected during the clinical observation of 4,000 patients. The detector was trained based on 6 types of seizures observed in the training data. However, when deployed, the CNN classifier may encounter patients who have types of seizures that do not exist in the training data because they are rare or even unknown by the medical community. Misclassifying these patients into the existing types of seizures brings serious risks and potential harm due to the potential for mistreatment of these patients. Ideally, in this case, the unknowns would be recognized and rejected by the classifier.

In this work, we focus on this important problem, describing a deep neural network that not only accurately classifies test objects into known target classes, but also correctly rejects unknowns.

**State-of-the-Art.** In a typical CNN, the output of the last fully connected layer is fed into a softmax layer to generate a class probability in $[0, 1]$ for each target class. An object will then be assigned to the class with the maximal probability. Intuitively, unknowns would be detected by leveraging this confidence, as was done in Bendale & Boult (2016); Hendrycks & Gimpel (2017); Liang et al. (2018). Since unknowns should not exhibit as many features of a target class versus known objects, the CNN should report a lower confidence. In prior work (Bendale & Boult, 2016; Hendrycks & Gimpel, 2017; Liang et al., 2018), the maximal probability or the largest value in the input vector to the softmax layer (maximal weighted sum) is used as a confidence to detect unknowns. In particular, an object will be rejected as an unknown if its confidence is smaller than a predetermined cutoff threshold $ct$.

However, as shown in our experiments (Sec. 5), these state-of-the-art methods are not particularly effective at rejecting unknowns. This is because CNNs achieve high classification accuracy by providing a strong ability to generalize, allowing it to overcome the gap between the training and testing data (Goodfellow et al., 2016). Unfortunately, this strength here is also a weakness, because it increases the chance of erroneously assigning an unknown to some target class even if it is quite different from the training objects in any target class. More specifically, the maximal probability (or maximal weighted sum) in a CNN is computed by the *weighted sum* operation on the multiple features produced by the convolutional layers. Because of this *sum* operation, an unknown can be classified to a target class with high confidence even if it matches some key features of a target class only by chance. Therefore, the requirements of accurately classifying the knowns and correctly rejecting the unknowns *conflict* with each other.

**Proposed Approach and Contributions.** In this work we propose an Unknown-aware Deep Neural Network (UDN for short) to overcome this problem.

The key intuition of UDN is to modify the CNN to use a *product* operation which models the *product relationship* among the features produced by the convolutional layers. This way, similar to the product rule in probability theory (Stroock, 2010), if just one feature indicative of a target class is not matched, the probability of assigning an object to this class is greatly reduced. Since an unknown is unlikely to match most of the features of a target class, the chance of assigning an unknown to a target class *with high confidence* is reduced. Therefore, the confidence produced by UDN should more effectively detect unknowns than the typical maximal probability/maximal weighted sum produced by classical CNNs. In UDN, the product operations are learned as a set of product relationship (PR) subnets leveraging the hierarchical nature of the binary tree structure. The strong bias of the classification decisions made via the product operations and the generalization ability introduced by the ensemble nature of multiple PR subsets together balance the contradictory requirements of accurately classifying known objects and correctly rejecting unknowns.

In addition, we propose an information-theoretic regularization strategy that actively incorporates the objective of unknown rejection into the learning process of UDN. This further improves the accuracy of UDN at rejecting unknowns by enlarging the confidence gap between unknown and known objects. We then show that the final loss function of UDN is fully differentiable. Therefore, UDN can be learned by following the common practice of back-propagation in deep neural networks.

We demonstrate the effectiveness of UDN using a rich variety of benchmark datasets including MNIST, CIFAR-10, CIFAR-100, and SVHN. UDN outperforms the state-of-the-art up to 20 points in the accuracy of unknown rejection – while preserving the accuracy of the underlying CNN at classifying objects from the target classes.

## 2   RELATED WORK

**Out-of-Distribution Detection.** In Bendale & Boult (2016), CNNs were adapted to discover unknowns by adding one additional unknown class to the softmax layer. This

method, called OpenMax, measures the distance between the maximal weighted sum vector produced for a test object and the mean maximal weighted sum vectors of the target classes that this test object is most likely assigned to. This distance is translated to a probability by the OpenMax layer. A testing object will be assigned to the unknown class if this probability is larger than a pre-defined cutoff threshold. Similarly, Malinin & Gales (2018) detects out-of-distribution objects by parameterizing a prior distribution over predictive distributions, so called Prior Networks (PNs), while Hendrycks & Gimpel (2017) directly uses softmax probabilities to detect out-of-distribution objects.

ODIN (Liang et al., 2018) detects unknowns using a two-pass inference strategy. That is, each test image goes through the inference stage twice. During the second inference round, each input is perturbed based on the gradient of its loss acquired in the first inference. The goal is to make the maximal probability produced by softmax more effective at separating unknowns. One problem with ODIN is that it introduces two extra hyper-parameters to control the level of perturbation, which are hard to tune.

As shown in experiments, our UDN method significantly outperforms OpenMax and ODIN, because UDN uses the maximal path probability as confidence measure, which as the product of multiple probabilities w.r.t. a set of nodes, is more effective in rejecting unknowns than using maximal weighted sum or maximal probability as a confidence measure.

MC-Dropout (Gal & Ghahramani, 2016) uses Bayesian model to reason about model uncertainty by casting dropout training in deep neural networks as as approximate Bayesian inference in deep Gaussian processes. It then rejects a testing object as unknown if the uncertainty about this object is large. As shown in experiments, MC-Dropout in many cases outperforms OpenMax and ODIN in rejecting unknowns, although is still worse than our UDN. Further, it scarifies the accuracy of classifying the known classes.

In Liu et al. (2018) and Vyas et al. (2018), methods were proposed to detect the objects that do not belong to known classes in a "clean" training dataset. However, these methods rely on a "contaminated" training set that contains a fraction of unknowns, i.e., that were not in the original clean training data. In other words, they solve the problem of rejecting "known unknowns", while our UDN instead focuses on the problem of rejecting "unknown unknowns". Our UDN does not require any labeled unknowns in training.

**Deep Outlier Detection.** Methods in Perera & Patel (2018); Erfani et al. (2016); Ruff et al. (2018); Nguyen & Vien (2018); Schlegl et al. (2017); Zhai et al. (2016); Chen et al. (2017); Zong et al. (2018); Hendrycks et al. (2019) use representation learning of deep neural networks to enhance the classical unsupervised outlier detection methods such as one-class classification (Schölkopf et al., 2001; Manevitz & Yousef, 2002; Tax & Duin, 2004) and neighbor-based methods (Breunig et al., 2000; Knorr & Ng, 1998; Ramaswamy et al., 2000; Bay & Schwabacher, 2003). However, unlike our work which focuses on enhancing the CNNs to reject the unknown objects during the inference process, these outlier detection methods do not take the classification objective into consideration. Instead, these methods detect outliers from a given dataset as the objects that significantly deviate from the majority of this dataset (Aggarwal, 2017), without using any labeled outliers or normal objects.

**Deep Neural Decision Forest.** Similar to the deep neural decision forest (DNDF) (Kontschieder et al., 2015), nodes in UDN at the final FC layers are connected to the split nodes of multiple trees. However, UDN is different from DNDF in several important ways. First, conceptually UDN corresponds to a variation of CNN model by replacing the softmax layer with tree structures such that the CNN can model the product relationship among the features learned by CNN, while DNDF instead enhances a random forest classifier with the feature learning ability of a CNN. Second, UDN incorporates the objective of unknown rejection into the learning process, while DNDF only considers the classification of known classes. Third, DNDF estimates both the decision node parametrizations $\theta$ and the leaf predictions $\pi$ by using a two-step optimization strategy, where $\theta$ and $\pi$ are updated alternatively to minimize the log-loss, while UDN is fully differentiable and therefore can be optimized in one step. Forth, in DNDF different trees share the FC layers (except the final FC layer) of the CNNs, while UDN divides all FC layers into $m$ independent components,

each of which is connected to one individual tree. This ensures independence among the trees and improves the classification accuracy because of the excellent generalization capability.

## 3 An Unknown-aware Deep Neural Network

In this section, we first introduce the structure of unknown-aware deep neural network (UDN). Next, we show how UDN distinguishes unknown from known objects. A regularization strategy is introduced in Sec. 4 to further improve the accuracy of UDN at rejecting unknowns.

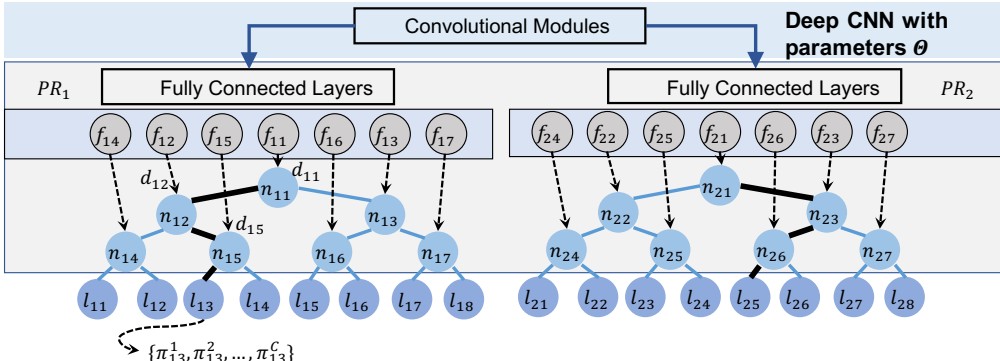

Figure 1: UDN Architecture.

### 3.1 UDN: Network Structure

As depicted in Fig. 1, UDN is composed of the convolutional module and $M$ independent product relationship ($PR$) subnets, where $M$ corresponds to a user definable hyper-parameter.

**PR Subnet.** The PR subnet is designed to model the *product relationship* among the features produced by the convolutional layers. Each PR subnet contains one fully connected (FC) component connected to a binary tree structure. Within each individual tree $T$, each split node of $T$ is connected to one output node of the final layer of the FC component. The mapping between the FC output nodes and the split nodes can be arbitrary. Any FC output node can be connected to the root node of the tree. The set of split nodes of $T$ is denoted as $\mathcal{N}$. The set of leaf nodes is denoted as $\mathcal{L}$.

Each split node $n_i \in \mathcal{N}$ converts the output $x_i$ of the FC node it consumes data from into a value in the $[0, 1]$ range by applying the sigmoid function $\sigma(x_i)$, where $\sigma(x_i) = (1 + e^{-x_i})^{-1}$.

Each leaf node $l \in \mathcal{L}$ is parameterized using a $C$-dimensional probability distribution $\pi_l$, where $C$ denotes the number of classes. One element $\pi_l^i$ of $\pi_l$ w.r.t. the $i$th class is computed as the softmax output of a *to be learned* parameter $w_l^i$:

$$\pi_l^i = softmax(w_l^i) = \frac{e^{w_l^i}}{\sum_{j=1}^{C} e^{w_l^j}} \tag{1}$$

**Product Relationship.** Next, we show how a PR subnet models the product relationship using the *path probability* $\mu_l(x)$ defined in Eq. 2, where $x$ denotes an input object and $l$ denotes a path from the root node to a leaf node $l$.

$$\mu_l(x) = \prod_{n_i \in \mathcal{N}_l} d(x_i) \tag{2}$$

In Eq. 2, $\mathcal{N}_l$ denotes the set of split nodes on the path from root to a leaf $l$. Given a node $n_i \in \mathcal{N}_l$, $d(x_i) = \sigma(x_i)$ if the *left child* of $n_i$ is included in this path. Otherwise, $d(x_i) = 1 - \sigma(x_i)$. For example, as shown in Fig. 1, on the path of $n_{11} \rightarrow n_{12} \rightarrow n_{15} \rightarrow l_{13}$, $d_{12}$ w.r.t. $n_{12}$ is set as $1 - \sigma(x_{12})$, because $n_{12}$ is connected to its right child $n_{15}$ on this path.

Therefore, $d(x_i)$ indicates the probability that input $x$ will be routed from node $n_i$ down to the next node on path $l$. Accordingly, $\mu_l(x)$ models the probability of input $x$ reaching leaf $l$, i.e., the probability of path $l$.

Since the $d(x_i)$ of each split $n_i$ corresponds to the output $x_i$ produced by the FC node w.r.t. $n_i$, essentially the path probability $\mu_l(x)$ is jointly determined by the output of multiple FC nodes. Therefore, $\mu_l(x)$ successfully models the *product relationship* among the features produced by CNN. The existence of one FC node that leads to small $d(x_i)$ will make the probability of the whole path $l$ small.

**Prediction.** At each leaf node $l$ of $T$, a PR subnet produces a prediction for a given input object $x$ using Eq. 3.

$$P_{PR}[y|x, \pi] = \sum_{l \in \mathcal{L}} \pi_{ly} \mu_l(x) \tag{3}$$

In Eq. 3, $\pi = \{\pi_l | l \in \mathcal{L}\}$, and $\pi_{ly}$ denotes the probability that leaf $l$ believes an input sample $x$ belongs to class $y$. $\mu_l(x)$ denotes the path probability of $l$. Generally speaking, the prediction is given by the probability produced at the leaf node $l$ weighted by the probability of input $x$ reaching leaf $l$.

Finally, as an ensemble of a set of PR subnets $\mathcal{PR} = \{PR_1, \ldots, PR_M\}$, UDN produces a prediction for an input $x$ by averaging the output of each subnet, i.e.:

$$P_{\mathcal{PR}}[y|x] = \frac{1}{|\mathcal{PR}|} \sum_{i=1}^{|\mathcal{PR}|} P_{PR_i}[y|x, \pi] \tag{4}$$

Similar to the typical ensemble structures like random forest (Ho, 1995), the ensemble of multiple PR subnets results in good generalization performance in classifying objects from target classes, even if each individual PR subnet could be overfit to the training examples.

Note when we setup the network structure of UDN, the mapping between the FC output nodes and the split nodes of the binary tree is arbitrary. The parameters w.r.t. each node is learned in an end-to-end fashion through back propagation. In the training process that minimizes the loss, the important features for a known class will be automatically learned and mapped to the split nodes on the same path. In other words, we do not have to group features explicitly. Instead, the grouping of the features is learned automatically. Fig. 1 shows the final results of the training. For example, since nodes $f_{11}$, $f_{12}$, and $f_16$ are not on the same path, these nodes are not considered to correspond to the key features of any class.

### 3.2 UDN: Unknown Rejection

**Max Path.** Based on the above architecture, given a testing object $x$, each PR subnet in UDN produces a probability distribution $\mu_l(x)$ over each path from the root to a leaf $l$. As shown in Eq. 2, the probability of a path is computed as the product of the probabilities $(d(x_n))$ produced by all split nodes on that path. Given an object $x$, a given path will have an extremely small probability if $x$ does not fit the features represented by the split nodes on this path (hence small $d(x_n)$), because the product of multiple small probabilities will diminish quickly. One path will stand out when all its split nodes produce large probabilities on $x$, marked as bold line in Figure 1. We call this path the *max path*, because it has the maximal probability among all paths.

Since the learned $\pi_l$ on each leaf is invariant w.r.t. input $x$, essentially it is the max path that determines the class of $x$ by Eq. 3. Therefore, the probability of the max path $\mu_l(x)$ (or *max path probability*) can be used to measure how confident the classifier is about its classification decision of object $x$. The larger the max path probability is, the more confident

the classifier is about the object. More specially, given an object $x$ and a PR subnet $PR$, the confidence is measured as:

$$CF_{PR}(x;\Theta) = max\{\mu_l(x)|l \in \mathcal{L}\} \tag{5}$$

where $max\{\mu_l(x)|l \in \mathcal{L}\}$ denotes the max path probability of subnet $PR$ for the given object $x$. Since UDN is an ensemble of a set of PR subnets, the final confidence of object $x$ is measured as:

$$CF_{\mathcal{PR}}(x) = \frac{1}{|\mathcal{PR}|} \sum_{i=1}^{|\mathcal{PR}|} CF_{PR_i}(x;\Theta_i) \tag{6}$$

**Effectiveness of Using Max Path to Reject Unknowns.** Intuitively this max path probability can be expected to be more effective at detecting unknowns than the maximal weighted sum in CNN. Typically an unknown will not get large probability on every split node on the max path, and one low probability node will limit the max path probability because of the product operation used in the computation. In contrast, the maximal weighted sum in CNN tends to fall off much more slowly because the score is computed based on the sum operation (with weights) on multiple features, such that a single matching feature can make the score high. This is confirmed in our experiments (Sec. 5.2, Appendix B, Appendix C).

## 4 Incorporating Unknown Rejection into the Learning Process

### 4.1 Information Theory-based Regularization

To ensure the effectiveness of using the max path probability to reject unknowns, we further incorporate the objective of unknown rejection into the learning process of UDN by introducing a regularization. The key idea is to use an information theory-based approach to prevent the generation of a PR subnet whose paths show uniform probability distribution. This ensures that the max path probability of each subnet will be generally much larger than the probabilities of other paths, making it more effective at rejecting unknowns. To achieve this, we penalize the paths whose probability distribution has a large entropy and hence is close to uniform (Pereyra et al., 2017).

Given a subnet $PR$ with $|\mathcal{L}|$ paths, each input $x \in \mathcal{X}$ can use any of $|\mathcal{L}|$ paths to reach leaf nodes. The entropy of the path probability distribution of input $x$ is given by:

$$H(\mu(x)) = -\sum_{l \in \mathcal{L}} \mu_l(x)log(\mu_l(x)) \tag{7}$$

We then apply a softmax function on the probability distribution as a normalization. The revised entropy of the path probability distribution is given by:

$$H(\mu'(x)) = -\sum_{l \in \mathcal{L}} \mu'_l(x)log(\mu'_l(x)) \tag{8}$$

where $\mu'_l(x) = softmax(\mu_l(x)) = \frac{exp(\mu_l(x))}{\sum_{l_i \in \mathcal{L}} exp(\mu_{l_i}(x))}$ is the softmax transformation of $\mu_l(x)$.

To penalize the subnet whose path probability distribution is close to uniform, we add the entropy w.r.t. each training sample to the log-loss term. Given the training set $\mathcal{X}$ and the output $\mathcal{Y}$, the penalized log-loss term of one subnet $PR$ is represented as:

$$L(PR;\mathcal{X},\mathcal{Y}) = \sum_{(x,y) \in \mathcal{X} \times \mathcal{Y}} -log(P_{PR}[y|x,\Theta,\pi]) + \beta H(\mu'(x|\Theta)) \tag{9}$$

where $\beta$ controls the strength of the penalty and $P_{PR}[y|x,\Theta,\pi]$ is defined in Eq. 3. $\Theta$ represents the learned parameters at the convolutional layers and FC layers.

The *total log-loss* for the UDN composed of $|\mathcal{PR}|$ subnets is then defined as:

$$L(\mathcal{PR}; \mathcal{X}, \mathcal{Y}) = \frac{1}{\mid \mathcal{PR} \mid} \sum_{i=1}^{|\mathcal{PR}|} L(PR_i; \mathcal{X}, \mathcal{Y}) \tag{10}$$

## 4.2 Training a UDN

Training a UDN requires finding a set of parameters $\Theta$ and $\pi$ that minimize the total log loss defined in Eq. 10. To minimize Eq. 10, we can independently minimize the penalized loss (Eq. 9) of each individual subnet. In Appendix A, we show that the loss function is fully differentiable. As a result, we are able to employ SGD to minimize the loss w.r.t. $\Theta$ and $\pi$, following the common practice of back-propagation in deep neural networks.

## 5 Experimental Evaluation

### 5.1 Overview of Experimental Setting

**Datasets.** We demonstrate the effectiveness of UDN on several benchmark image datasets. Specifically, we train models on CIFAR-10 (Krizhevsky & Hinton, 2009), CIFAR-100 (Krizhevsky, 2009), and SVHN (Netzer et al., 2011) datasets. Given a trained model on one dataset, we consider examples from other datasets as unknowns when testing the model. In addition, we also sample some classes from the training data as unknowns and test these samples on the model trained for the rest of the classes. Due to the lenght constraints we present the results on the CIFAR-100 and SVHN models in Appendix B and Appendix C.

**Methodology.** We evaluate: (1) CNNs with *Weighted-Sum* as a baseline. The weighted sum score is utilized as the confidence measure; (2) OpenMax (Bendale & Boult, 2016), (3) ODIN (Liang et al., 2018), (4) *Softmax* (Hendrycks & Gimpel, 2017), and (5) MC-Dropout (Gal & Ghahramani, 2016): the state-of-the-art unknown rejection methods described in related work (Sec. 2); (6) our *UDN* model without the regularization term applied to the loss function and (7) *UDN-Penalty*: our UDN with the regularization term (Eq. 9).

The results show that our UDN and UDN-Penalty significantly outperform OpenMax, ODIN, MC-Dropout and *Softmax* through a variety of unknown rejection experiments, while still preserving the *accuracy of classifying* objects from target classes.

**Experimental Setup.** We ran experiments on 4 P100 GPU instances on Google cloud. All models are implemented in Pytorch (Paszke et al., 2017).

**Hyper-parameter Settings.** All networks are trained using mini-batches of size 128. The momentum is set to 0.9 for all models. The weighted decays are set to 0.0001. When testing MNIST on models trained for other datasets, we increase its color channel from 1 to 3 by copying the original gray images 3 times.

**Evaluation Metric.** Following the literature of unknown rejection, we use two metrics to measure the effectiveness of UDN at distinguishing known and unknown images, namely true negative rate (TNR) at 95% true positive rate (TPR) and AUROC. TNR at 95% TPR can be interpreted as the probability that an unknown image (negative) is correctly recognized when the TPR is 95%. True positive rate can be computed by TPR = TP /(TP + FN), where TP and FN denote true positives (knowns are correctly classified as knowns) and false negatives (known images are misclassified as unknowns) respectively. The true negative rate (TNR) can be computed by TNR = TN/(FP+TN), where FP and TN denote false positives (unknowns are misclassified as knowns) and true negatives (unknowns are correctly recognized) respectively.

AUROC corresponds to the Area Under the Receiver Operating Characteristic curve, which is a threshold independent metric (Davis & Goadrich, 2006). It can be interpreted as the probability that a positive example is assigned a higher detection score than a negative example. In addition, we also measure the accuracy of these methods at classifying the knowns into target classes.

## 5.2 CIFAR-10

We tested all approaches on DenseNet (Huang et al., 2017). All methods use the same DenseNet architecture to ODIN (Liang et al., 2018). For our UDN and UDN-Penalty, the output is connected to 10 depth-5 trees. Specifically, the output of the convolutional layer is broadcast to 10 different sets of FC layers. Each set contains 3 FC layers. The final FC layer with 63 ($2^{(5+1)} - 1$) hidden nodes is connected to a tree. The training time of UDN is 9.4 hours, slightly slower than training a DenseNet model (9.1 hour). For the evaluation of ODIN, we directly use the model published by the authors. The temperature parameter $T$ and the perturbation magnitude $\eta$ used by ODIN are set to 1000 and 0.0014. We set the drop rate of MC-Dropout as 0.2 and the number of forward passes as 100. We set these parameters by the suggestion of the authors or parameter tuning.

Table 1: CIFAR-10 Results (DenseNet).

| Methods | TNR (95% TPR) | | | AUROC | | | Classification Accuracy |
|---|---|---|---|---|---|---|---|
| | CIFAR-100 | SVHN | MNIST | CIFAR-100 | SVHN | MNIST | |
| Weighted-Sum (Baseline) | 52.25% | 54.96% | 98.96% | 90.33% | 91.80% | 99.51% | 94.26% |
| Softmax (Hendrycks & Gimpel, 2017) | 40.35% | 41.64% | 76.38% | 89.03% | 90.54% | 96.92% | 94.26% |
| OpenMax (Bendale & Boult, 2016) | 51.97% | 63.67% | 49.56% | 90.54% | 93.28% | 89.73% | 94.26% |
| ODIN (Liang et al., 2018) | 50.8% | 49.57% | 99.14% | 89.80% | 91.24% | 99.86% | 95.19% |
| MC-Dropout (Gal & Ghahramani, 2016) | 51.86% | 78.73% | **99.82**% | 78.91% | 89.5% | **99.94**% | 93.95% |
| UDN | 54.22% | 84.64% | 98.98% | 92.93% | 97.59% | 98.82% | **95.30**% |
| UDN-Penalty | **56.55**% | **88.09**% | 99.07% | **93.47**% | **98.71**% | 99.77% | 94.27% |

As shown in Table 1, in almost all cases UDN and UDN-Penalty outperform Softmax, Weighted-sum, OpenMax, ODIN, and MC-Dropout in rejecting unknowns, without giving up our ability to correctly classify the CIFAR-10 images. Specifically, UDN and UDN-Penalty outperforms other methods up to 10 points in TNR. UDN and UDN-Penalty also significantly outperform other methods in AUROC. In particular, UDN and UDN-Penalty achieve about 98% AUROC when detecting SVHN as unknowns. This indicates that our methods can effectively separate knowns and unknowns under a wide range of parameter settings.

The performance gain results from our UDN architecture, where the confidence of each image is computed as the product of the probabilities produced at the split nodes on the path. This multiplication of probabilities enlarges the confidence gap, making it able to better reject unknowns than alternate approaches. The only exception is that MC-Dropout performs the best on rejecting MNIST as unknowns, showing that MC-Dropout is probably good at separating unknowns which are simplistic yet very different from known objects.

In addition, UDN achieves slightly better classification accuracy compared to Softmax, Weighted-Sum, OpenMax, and ODIN that use the classical DenseNet. The classification accuracy of MC-Dropout is worse than other methods because of the Bayesian inference.

Our UDN-Penalty outperforms UDN in rejecting unknowns in all cases. This is because by introducing regularization to penalize path probability distributions that have a large entropy, UDN-Penalty leads to larger maximal path probabilities for inliers. At the same time, its classification accuracy decreases slightly, because the regularization introduces overfitting (Szegedy et al., 2016). However, this is effectively alleviated because of the ensemble of the PR subnets. Therefore, the drop in classification accuracy is very small.

## 6 CONCLUSION

In this work, we proposed an augmentation to CNNs, UDN, which effectively rejects unknown objects that do not belong to any class seen in the training data. UDN achieves this by replacing softmax layer in traditional CNNs with a novel tree ensemble that takes the product of feature values, balancing the contradictory requirements of accurately classifying knowns and correctly rejecting unknowns in one network structure. A regularization strategy is proposed for UDN to further enhance its unknown rejection capacity.

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

## A  LEARNING OF UDN

**Learning Decision Nodes by Back-Propagation**

Given a decision tree, the gradient of the loss $L$ with respect to $\Theta$ can be decomposed by the chain rule as follows:

$$\frac{\partial L}{\partial \Theta}(\Theta, \pi; x, y) = \sum_{n \in \mathcal{N}} \frac{\partial L(\Theta, \pi; x, y)}{\partial f_n(x; \Theta)} \frac{\partial f_n(x; \Theta)}{\partial \Theta} \tag{11}$$

Here, the derivative of the second part $\frac{\partial f_n(x; \Theta)}{\partial \Theta}$ is identical to the back-propagation process of traditional CNN modules and thus is omit here. Now let's show how to compute the first part:

$$\begin{aligned} \frac{\partial L(\Theta, \pi; x, y)}{\partial f_n(x; \Theta)} &= \frac{\partial(-log(P_T[y|x, \Theta, \pi]) + \beta H(\mu'(x|\Theta)))}{\partial f_n(x; \Theta)} \\ &= \frac{\partial(-log(P_T[y|x, \Theta, \pi])}{\partial f_n(x; \Theta)} + \beta \frac{\partial(H(\mu'(x|\Theta)))}{\partial f_n(x; \Theta))} \end{aligned} \tag{12}$$

where

$$\frac{\partial(-log(P_T[y|x, \Theta, \pi])}{\partial f_n(x; \Theta)} = -\sum_{l \in \mathcal{L}} \frac{\pi_{ly}}{\mathbb{P}_T[y|x, \Theta, \pi]} \frac{\partial \mu_l(x|\Theta)}{\partial f_n(x; \Theta)} \tag{13}$$

and

$$\begin{aligned} \sum_{l \in \mathcal{L}} \frac{\partial \mu_l(x|\Theta)}{\partial f_n(x; \Theta)} &= -\sum_{l \in \mathcal{L}} \mu_l(x|\Theta) \frac{\partial log(\mu_l(x|\Theta))}{\partial f_n(x; \Theta)} \\ &= -\sum_{l \in \mathcal{L}} \mu_l(x|\Theta)(\mathbb{1}_{l \swarrow n} \bar{d}_n(x; \Theta) - \mathbb{1}_{l \searrow n} d_n(x; \Theta)) \\ &= -\sum_{l \in \mathcal{L}_{n_l}} \mu_l(x|\Theta) \bar{d}_n(x; \Theta) + \sum_{l \in \mathcal{L}_{n_r}} \mu_l(x|\Theta) d_n(x; \Theta) \end{aligned} \tag{14}$$

By using the chain rule, we get:

$$\frac{\partial(H(\mu'(x|\Theta)))}{\partial f_n(x; \Theta))} = \sum_{l \in \mathcal{L}} \frac{\partial(H(\mu'(x|\Theta)))}{\partial(\mu'_l(x|\Theta))} \frac{\partial(\mu'_l(x|\Theta))}{\partial(\mu_l(x|\Theta))} \frac{\partial(\mu_l(x|\Theta))}{\partial f_n(x; \Theta))} \tag{15}$$

where

$$\begin{aligned} \sum_{l \in \mathcal{L}} \frac{\partial(H(\mu'(x|\Theta)))}{\partial(\mu'_l(x|\Theta))} &= \sum_{l \in \mathcal{L}} \frac{\partial(\mu'_l(x|\Theta) log(\mu'_l(x|\Theta)))}{\partial(\mu'_l(x|\Theta))} \\ &= \sum_{l \in \mathcal{L}} (1 + log(\mu'_l(x|\Theta))) \end{aligned} \tag{16}$$

and

$$\sum_{l \in \mathcal{L}} \frac{\partial(\mu'_l(x|\Theta))}{\partial(\mu_l(x|\Theta))} = \sum_{l \in \mathcal{L}} (s_l(1 - s_l) + \sum_{k \neq l} s_l s_k)$$

where

$$s_l = \frac{e^{\mu_l(x|\Theta)}}{\sum_{k \in \mathcal{L}} e^{\mu_k(x|\Theta)}}$$

**Learning Prediction Nodes by Back-Propagation**

Given a decision tree, the gradient of the Loss $L$ w.r.t. the weights $w$ of the prediction nodes (defined in Eq. 1) can be decomposed by the chain rule as follows:

$$\frac{\partial L}{\partial w}(\Theta, \pi; x, y) = \sum_{l \in \mathcal{L}} \frac{\partial L(\Theta, \pi; x, y)}{\partial \pi_l} \frac{\partial \pi_l}{\partial w_l} \tag{17}$$

where

$$\frac{\partial L(\Theta, \pi; x, y)}{\partial \pi_{ly}} = -\frac{\mu_l(x|\Theta)}{P_T[y|x, \Theta, \pi])} \tag{18}$$

and

$$\frac{\partial \pi_{ly}}{\partial w_{li}} = \begin{cases} \pi_{ly}(1 - \pi_{ly}) & y = i \\ -\pi_{ly}\pi_{li} & y \neq i \end{cases} \tag{19}$$

Therefore,

$$
\begin{aligned}
\frac{\partial L(\Theta, \pi; x, y)}{\partial w_{li}} &= \sum_{y \in \mathcal{Y}} \frac{\partial L(\Theta, \pi; x, y)}{\partial \pi_{ly}} \frac{\partial \pi_{ly}}{\partial w_{li}} \\
&= \frac{\partial L(\Theta, \pi; x, i)}{\partial \pi_{li}} \frac{\partial \pi_{li}}{\partial w_{li}} + \sum_{y \neq i} \frac{\partial L(\Theta, \pi; x, y)}{\partial \pi_{ly}} \frac{\partial \pi_{ly}}{\partial w_{li}} \\
&= -\frac{\mu_l(x|\Theta)(\pi_{li}(1 - \pi_{li}))}{P_T[i|x, \Theta, \pi])} + \sum_{y \neq i} \frac{\mu_l(x|\Theta)\pi_{ly}\pi_{li}}{P_T[y|x, \Theta, \pi])}
\end{aligned}
\tag{20}
$$

Table 2: CIFAR-100 Results (DenseNet).

| Methods | TNR (80% TPR) | | | AUROC | | | Classification Accuracy |
|---|---|---|---|---|---|---|---|
| | CIFAR-10 | SVHN | CIFAR-100 Classes | CIFAR-10 | SVHN | CIFAR-100 Class | |
| Weighted-Sum (Baseline) | 53.70% | 74.84% | 50.65% | 74.54% | 81.61% | 74.22% | 77.1% |
| Softmax (Hendrycks & Gimpel, 2017) | 49.95% | 60.44% | 48.38% | 75.61% | 81.4% | 74.16% | 77.1% |
| OpenMax (Bendale & Boult, 2016) | 32.86% | 57.23% | 30.06% | 58.46% | 71.59% | 57.74% | 77.1% |
| ODIN (Liang et al., 2018) | 50.29% | 64.99% | 49.01% | 73.06% | 82% | 72.89% | 77.1% |
| MC-Dropout (Gal & Ghahramani, 2016) | 33.64% | 59.57% | 31.28% | 59.43% | 73.09% | 58.37% | 73.74% |
| UDN | 59.59% | 83.52% | 59.21% | 79.93% | 87.42% | 79.67% | 76.92% |
| UDN-Penalty | 59.92% | 84.55% | 59.67% | 82.03% | 87.59% | 81.81% | 76.77% |

## B CIFAR-100 EXPERIMENTS

Similar to the CIFAR-10 experiments, all methods use the same DenseNet (Huang et al., 2017) architecture to ODIN (Liang et al., 2018). For our UDN and UDN-Penalty, the output is connected to 10 trees. The depth of each tree is 6. Again, for the evaluation of ODIN, we directly use the model published by the authors. The temperature parameter $T$ and the perturbation magnitude $\eta$ used by ODIN are set to 1000 and 0.0014, as recommended by the authors. We set the drop rate of MC-Dropout as 0.2 after parameter tuning and set the number of forward passes as 100 as suggested by the authors.

In this set of experiments we use the images in CIFAR-10 and SVHN as the unknowns. Moreover, we also randomly pick 10 classes from the CIFAR-100 training data. We then use these samples as unknowns and test them on the model trained using the rest of the

CIFAR-100 training data. We run this process for 10 times and report the average TNR and AUROC.

As shown in Table 2, our UDN significantly outperform other methods in all cases by at least 9 points in TNR and 5 points in AUROC.

## C  SVHN EXPERIMENTS

Table 3: SVHN Results (DenseNet).

| Methods | TNR (95% TPR) | | | AUROC | | | Classification Accuracy |
|---|---|---|---|---|---|---|---|
| | CIFAR-10 | CIFAR-100 | SVHN-Classes | CIFAR-10 | CIFAR-100 | SVHN-Class | |
| Weighted-Sum (Baseline) | 80.76% | 77.83% | 60.92% | 95.70% | 94.64% | 92.22% | 96.42% |
| Softmax (Hendrycks & Gimpel, 2017) | 70.08% | 68.82% | 49.13% | 95.06% | 94.51% | 90.24% | 96.42% |
| OpenMax (Bendale & Boult, 2016) | 62.66% | 62.845% | 33.32% | 86.78% | 85.65% | 59.78% | 96.42% |
| ODIN (Liang et al., 2018) | 79.32% | 76.01% | 61.56% | 94.87% | 93.52% | 92.11% | 96.42% |
| MC-Dropout (Gal & Ghahramani, 2016) | 57.41% | 55.25% | 58.62% | 78.19% | 77.24% | 79.15% | 92.94% |
| UDN | 96.33% | 92.10% | 74.06% | 98.96% | 98.15% | 94.82% | 96.577% |
| UDN-Penalty | 96.50% | 92.66% | 74.08% | 98.94% | 98.22% | 95.10% | 96.543% |

Same to the CIFAR-100 experiments above, we tested all approaches based on the same DenseNet architecture. For our UDN and UDN-Penalty, the output is connected to 10 trees. The depth of each tree is 4. For ODIN, the temperature parameter $T$ and the perturbation magnitude $\eta$ used by ODIN are set to 1000 and 0.0014 after parameter tuning. We set the drop rate of MC-Dropout as 0.5 and the number of forward passes as 100 as suggested by the authors.

We use the images in CIFAR-10 and CIFAR-100 as the unknowns. At the same time we also randomly pick 1 classes from the SVHN training data and use these samples as unknowns. We then test them on the model trained using the rest of the SVHN training data. We run this process for 10 times and report the average TNR and AUROC.

As shown in Table 3, in all cases our UDN outperform other methods in both TNR and AUROC. In particular, UDN outperforms all other methods by at least 12 points in TNR.

## D  PARAMETER TUNING

In our UDN, The key hyper-parameters are the number of PR subnets and the depth of the binary tree. In general, the depth of the tree depends on the number of the classes. The more classes the dataset has, the higher the tree should be. In our experiments, when the depth of the tree is no smaller than 6 and the number of PR subnet is above 10, our method works well in general. When the number of classes in the training data is small such as MINIST and CIFAR-10, setting the depth of the three as a smaller value such as 4 also works. But a larger depth of the tree or more subnets does not harm the effectiveness of our UDN method. Therefore, these parameters are easy to tune.

