# OpenReview forum: "Unknown-Aware Deep Neural Network"
_ICLR.cc/2020/Conference — Reject_

### Official Review · AnonReviewer2 · 2019-10-22
**Official Blind Review #2**

**Rating:** 8

**Review:**

This paper proposes a neural network architecture for image classification, which can more accurately recognize the unknown class that is not presented in the training data than the prior work. The key idea is to organize the features into a binary tree and use the product of probabilities along the paths to the leaf node to predict whether the test image has all the relevant features that should present in known classes. This proposed method is compared to multiple baselines and demonstrates superior results in image classification, especially for correctly predicting the unknown class.

I am not an expert in image classification. Thus I cannot judge based on the novelty of this paper. I vote for acceptance solely based on the clear writing, technical soundness, thorough evaluation and good results. The intuition behind the method that missing a key feature of a particular class should significantly reduce the probability of assigning the input to this class is well captured by the product of probabilities along a path in a binary tree structure. In other words, the proposed method makes sense intuitively and is validated on several datasets. One small concern that I have is that a binary tree explicitly partitions the feature into separated groups. For example, if for a known class, f11, f12 and f16 are the three important features to be present. According to Figure 1, there is no path that connects all these three nodes. Will different partitioning of the features affect the accuracy of the classification, including the unknown class?

To improve the paper, it would be great to add one more experiment to examine the effect of the ensemble, for example, by varying the number of product relationship module.

-------------------------Update after rebuttal-------------------------
Thanks for your detailed response and the additional experiments. The response addressed my questions and concerns. Thus I will keep my original recommendation of acceptance.

**Experience Assessment:**

I do not know much about this area.

**Review Assessment: Checking Correctness Of Derivations And Theory:**

I assessed the sensibility of the derivations and theory.

**Review Assessment: Checking Correctness Of Experiments:**

I assessed the sensibility of the experiments.

**Review Assessment: Thoroughness In Paper Reading:**

I read the paper at least twice and used my best judgement in assessing the paper.

---

> ### Author Response · Authors · 2019-11-15
> **Response to Reviewer 2: The problem of feature partitioning**
>
> We thank the reviewer for the positive review. Here we respond to your small concern and suggestion.
>
> Small concern: A binary tree explicitly partitions the feature into separated groups. For example, if for a known class, f11, f12 and f16 are the three important features to be present. According to Figure 1, there is no path that connects all these three nodes. Will different partitioning of the features affect the accuracy?
>
> Response: When we setup the network structure of our UDN, each split node of the binary tree is connected to one output node of the final layer of the FC component. The mapping between the FC output nodes and the split nodes is arbitrary. Then the parameters with respect to each node is learned in an end-to-end fashion through back propagation. In the training process, to minimize the loss, the important features for a known class will be automatically learned and mapped to the split nodes on the same path. In other words, we don't partition features explicitly. Instead, the partitioning of the features is learned automatically. Figure 1 just shows the final results of the training process. If f11, f12 and f16 are not on the same path, f11, f12 and f16 are not considered to correspond to the key features of any class.
>
> We have modified our paper to reflect this. Thank you for the careful thinking. We believe this improves the readability of our work.
>
> Suggestion: The effect of the number of product relationship modules,
>
> Response: Based on our current experiments, we can get very good results after the number of product relationship modules increases to 10. After that, using more product relationship modules does not clearly help. We will do a more thorough investigation of this in the future.

---

### Official Review · AnonReviewer3 · 2019-10-22
**Official Blind Review #3**

**Rating:** 3

**Review:**


This paper proposes the unknown-aware deep neural network (UDN), which can be used to discover out-of-distribution samples for neural network classifiers. Its main idea is to introduce PR subnets to model the product relationship instead of the dot product of regular networks, then it can avoid over-fitting. Experimental results demonstrate that UDN can discover unknown samples more precisely than several baselines.

The problem of learning with out-of-distribution samples is important for real-world applications. The results provided in this paper seem positive. However, I think the main idea is not well explained, and the experiments provided are not sufficient. It’s not clear why introducing this PR subnet forest can help the model avoid overfitting, and why this structure is more beneficial than other simple ensemble structures. I think it’s more helpful to provide additional insights or theoretical analysis. As for the experiments, the contents of images in CIFAR, SVHN, and MNIST are completely different, so under the given setting, the unknown samples are easy to detect. It’s more practical and convincing to split categories in each dataset into two parts to simulate unknowns.
Below are some detailed comments:
-      Can you compare the performance of UDN with Bayesian neural networks, since BNNs are also popular methods to model uncertainty?
-      How does the hyper-parameters for PR subnets affect the results?


**Experience Assessment:**

I have read many papers in this area.

**Review Assessment: Checking Correctness Of Derivations And Theory:**

I assessed the sensibility of the derivations and theory.

**Review Assessment: Checking Correctness Of Experiments:**

I assessed the sensibility of the experiments.

**Review Assessment: Thoroughness In Paper Reading:**

I read the paper at least twice and used my best judgement in assessing the paper.

---

> ### Author Response · Authors · 2019-11-15
> **Response to Reviewer 3: The novelty and the new experiments**
>
> Thanks for the insightful comments. Below we summarize and response to all of them.
>
> Comment 1: It’s not clear why introducing this PR subnet forest can help the model avoid overfitting, and why this structure is more beneficial than other simple ensemble structures.
>
> Response: Here we want to clarify that the main purpose of introducing the concept of PR subnet is to model the product relationship among the features produced by the convolutional layers. In this way, missing a key feature of a particular class should significantly reduce the probability of assigning an unknown object to this class, and therefore it is more effective in rejecting the unknown objects. This corresponds to our key technical novelty.
>
> The ensemble of multiple PR subnets is able to enhance the generalization capability of the model, similar to typical ensemble structures such as random forests. We are not inventing new techniques in this sense. We have modified our paper to make this more explicit.
>
> Comment 2: Compare the performance of UDN with Bayesian neural networks, since BNNs are also popular methods to model uncertainty.
>
> Response: Thank you for this valuable suggestion. In our new experiments we have compared against MC-Dropout, a popular BNN method. In deed, in many cases it works better than other state-of-the-art methods in rejecting unknowns. However, our UDN still consistently outperforms MC-Dropout. The only cases that MC-Dropout works slightly better than our UDN are the experiments of rejecting MNIST as unknowns from the CIFAR-10 model. Since MNIST is very simple and significantly different from CIFAR-10, these are easy cases for all methods. For the results, please refer to Table 1 (Sec. 5.2), and Table 2 (Appendix B), and Table 3 (Appendix C).
>
> Comment 3: As for the experiments, the contents of images in CIFAR, SVHN, and MNIST are completely different, the unknown samples are easy to detect. It’s more practical and convincing to split categories in each dataset into two parts to simulate unknowns.
>
> Response: Indeed it is relatively easy to detect the unknowns if the unknown are completely different from the training data. It makes more sense if a method can detect the unknowns categories that are close to the training categories. Therefore, per the suggestion, we pick some categories from the training data itself to simulate unknown when we run the new experiments on the CIFAR-100 and SVHN models per the request of Reviewer 1. As shown in Tables 2 and 3 (Appendix A and Appendix B), in all cases our UDN method consistently outperforms all other methods including the BNN method MC-Dropout at least 9 percentage points in the accuracy of rejecting unknowns.
>
> Note that since CIFAR-10 and CIFAR-100 are extracted from the same source (ImageNet), have the same resolution and relevant categories, the CIFAR-10 and CIFAR-100 datasets have very similar properties. Therefore, our experiments that use CIFAR-100 as the unknowns to test the CIFAR-10 model or vice versa also confirm that our UDN is effective in capturing the unknowns that are similar to the training data.
>
> Comment 4: How does the hyper-parameters for PR subnets affect the results?
>
> Response: The key hyper-parameters for PR subnets are the depth of the binary tree and the number of PR subnets. In general, the depth of the tree depends on the number of the classes. The more classes the dataset has, the higher the tree should be. In our experiments, when the depth of the tree is no smaller than 6 and the number of PR subnet is above 10, our method works well in general. When the number of classes in the training data is small such as MINIST and CIFAR-10, setting the depth of the three as a smaller value such as 4 or 5 also works. But a larger depth of the tree or more subnets does not harm the effectiveness of our UDN method. Therefore, these parameters are easy to tune. The above description has also been added into our paper Appendix D.

---

### Official Review · AnonReviewer1 · 2019-10-27
**Official Blind Review #1**

**Rating:** 3

**Review:**

This paper is about a novel method to detect unknown samples which are of a different class than the trained ones. The idea is to use an output subnet which use a fully connected layer and a binary tree which encode the product relationship instead of the sum currently used in state of the art method (particularly the softmax with low confidence). The binary tree is made of split nodes which are responsible to produce a probability distribution from the root to each leaf. The max path i.e. the path with the largest probabilities determines the class of the input and can be used to measure how confident the classifier is about the classification decision. Combine multiple subnets and the tool obtained is able to do complex predictions and maintain a good generalization performance. The method also uses an information theory based regularization which decrease the probability of having subnets with uniform probability distribution i.e. a large entropy. Experiments on CIFAR-10 and MNIST against CIFAR-100, SVHN show that the method has an improved rejection accuracy while maintaining a good classification accuracy on the test set.

The topic of the paper is interesting and the approach seems to be solid, however the experiments are not so convincing. They are limited on two very easy datasets and do not show if the method is able to scale when more difficult and more realistic amount of classes are considered (like in CIFAR-100, SVHN or maybe ImageNet). Given that a simple dataset like these require 9 hour of training, it is also not clear how much the method is able to scale computationally and if it is applicable realistically. Moreover the presentation could be improved as figure 1 and its section 2 are complex and not easy to follow at several points. Hence, I’m leaning towards rejecting this paper.

In particular:
- It would be interesting to see experiments where the number of classes is higher than the ones in MNIST and CIFAR-10. CIFAR-100 and SVHN would be a good testbed for such case.
- How is the complexity of the method and how does it scale with the number of training classes?
- It is stated that the method bring a 25% percentage points in the accuracy of unknown rejection detection, however table 1 shows a large improvement only in the case of SVHN. Hence the claims seems a bit off compared to the measured data. Moreover, using only CIFAR-10 is insufficient to back up the claim.
- Why the related work section is at the end of the paper? It confuses the reader and would be more useful to be put after the introduction section.



**Experience Assessment:**

I have read many papers in this area.

**Review Assessment: Checking Correctness Of Derivations And Theory:**

I assessed the sensibility of the derivations and theory.

**Review Assessment: Checking Correctness Of Experiments:**

I carefully checked the experiments.

**Review Assessment: Thoroughness In Paper Reading:**

I read the paper at least twice and used my best judgement in assessing the paper.

---

> ### Author Response · Authors · 2019-11-15
> **Response to Reviewer 1: New experiments show the advantage of our method on complex data**
>
> We thank the reviewer for the valuable comments. We summarize and number the comments and response to each of them.
>
> Comment 1: It would be interesting to see experiments where the number of classes is higher than the ones in MNIST and CIFAR-10. CIFAR-100 and SVHN would be a good testbed for such case.
>
> Response: Per the suggestion of the reviewer, we ran additional experiments on the CIFAR-100 and SVHN datasets. Compared to the other methods, in all cases our UDN method achieves at least 9 percentage point gains in detection accuracy as shown in Tables 2 and 3 (Appendix B and Appendix C). In the new experiments, due to time constraints, we did not test the case of using MNIST as unknown objects, because MNIST is a simple dataset and is easy to separate out.
>
> Comment 2: How is the complexity of the method and how does it scale with the number of training classes?
>
> Response: Our method is not much more complex than the traditional CNN network architecture. As confirmed in our experiments, it took about 9 hours to train a DenseNet model on CIFAR-10 data set, where DenseNet is well known to be able to get high classification accuracy on CIFAR-10, While it took about 9.5 hours to train our UDN model -- only slightly slower than training the DenseNet model.
>
> As confirmed in the new CIFAR-100 experiments we ran, our UND works well and outperforms the other approaches more when the number of training classes is large. As shown in Table 2 (Appendix B), our UDN outperforms all other methods by at least 9 percentage points in terms of the accuracy of rejecting unknowns.
>
> Comment 3: The 25 percentage point gains only happy in the case of SVHN.
>
> Response: We have modified our paper accordingly and clearly show this happens only at the best case.
>
> Comment 4: Why the related work section is at the end of the paper?
>
> Response: In the revised paper, we have put the related work section after the introduction.
>
> Comment 5: The presentation could be improved as Figure 1 and its Section 2 are complex.
>
> Response: Per the suggestion, we have modified Section 2 to make it clearer. In particular, at the end of Section 2, we discuss Figure 1 more and explain why the network structure of UDN can be designed like this.

---

### Decision · Program_Chairs · 2019-12-19

**Decision:**

Reject

**Comment:**

This paper proposes the unknown-aware deep neural network (UDN), which can discover out-of-distribution samples for CNN classifiers. Experiments show that the proposed method has an improved rejection accuracy while maintaining a good classification accuracy on the test set. Three reviewers have split reviews. Reviewer #2 provides positive review for this work, while indicating that he is not an expert in image classification. Reviewer #1 agrees that the topic is interesting, yet the experiment is not so convincing, especially with limited and simple databases. Reviewer #3 shared the similar concern that the experiments are not sufficient. Further, R3 felt that the main idea is not well explained. The ACs concur these major concerns and agree that the paper can not be accepted at its current state.